# Design and Experiment of a Helicoidal Seed Tube to Improve Seed Distribution Uniformity of Seed Drills

Davut Karayel [1,2,*], Egidijus Šarauskis [2] and Ali Aktaş [1]

1   Department of Agricultural Machinery and Technologies Engineering, Faculty of Agriculture,
    Akdeniz University, Antalya 07058, Turkey; aliactas@gmail.com
2   Agriculture Academy, Department of Agricultural Engineering and Safety, Vytautas Magnus University,
    LT-46299 Kaunas, Lithuania; egidijus.sarauskis@vdu.lt
*   Correspondence: dkarayel@akdeniz.edu.tr

**Abstract:** In a conventional seed drill, the flow of seeds into the planting bed is usually disrupted and irregular, which contributes to poor seed spacing uniformity. The seed tube of the seed drill may be one of the reasons for disrupted and irregular seed flow. A seed drill, in principle, has to deposit seeds into the ground at a uniform spacing to avoid nutrient competition for optimum yield. This study was inspired to develop a novel helicoidal seed tube that regulates the flow of seeds in a seed tube while being dispensed by the seed drill into the ground. The helicoidal tube was designed to fit the end of a conventional delivery tube of a seed drill. It works by allowing the falling seeds to move through the helix causing a frictional reaction, thereby regulating the velocity of the falling seeds. The developed helicoidal seed tube was tested under laboratory conditions at three heights of the helix (100, 150, and 200 mm) and four pitch sizes (28, 32, 36, and 40 mm). As a result of the laboratory tests, the optimum values for the helix height and pitch size were determined as 200 mm and 36 mm, respectively. The performance of the helicoidal seed tube (helix height of 200 mm and pitch size of 36 mm) was compared with the conventional (hollow) seed tube under laboratory and field conditions. Trials were carried out at a seeding rate of 200 kg/ha and a forward speed of 1 m/s for both wheat and barley seeds. Field trials showed that the coefficients of variation of seed spacing of the conventional seed tube decreased from 118.4% and 139.5% to 77.2% and 70.6% for wheat and barley sowing, respectively, when the helicoidal seed tube was used. Overall, the use of helicoidal seed tube provided a more equal growing area for each plant due to its lower coefficient of variation.

**Keywords:** seeder; seed tube; helix; seed spacing; seed flow; wheat; barley





## 1. Introduction

The sowing uniformity of a seed drill is a critical factor affecting the emergence, plant development and yield. In addition to factors that increase crop yield, such as tillage, fertilization, and plant protection, an increase in crop yield can be achieved by providing a uniform seed distribution and optimum growing area. After the sowing process, the distribution of seeds in the soil is expressed as a horizontal and vertical distribution. Metering mechanisms of seed drills are primarily effective in ensuring a uniform seed distribution in a horizontal plane. But it should be taken into consideration that the problems that will occur until the seeds, which are activated (delivered) by the metering mechanism, are placed in the soil have an effect on seed distribution uniformity. The seed distribution uniformity in the horizontal plane is influenced by the distance between the successive seeds as well as the deviation of seeds from the furrow axis. The distance between the row and seed spacings in the row determines the plant growing area of seeds. The deviations from the theoretical spacing constitute the seed distribution uniformity in the horizontal plane. Improving the seed distribution uniformity of a seed drill will allow a seed to have its optimum growing area that will avoid the competition for nutrients with its neighboring plants, which in turn increases yield [1–3].

The improvement of the seed distribution uniformity in the sowing process contributes to the proximity of the plant growing area per seed relative to each other and thus to the increase of yield [1]. In addition, the reduction of weeds is achieved by reducing the gaps between neighboring seeds. Generally, the seed distribution uniformity in the horizontal plane and the distribution uniformity of the plant growing areas are not very good in conventional seed drills. According to Müller and Köller, mechanical seed drills distribute seeds exponentially. The rate of seeds sown at the desired spacing over the row was very low for mechanical seed drills. The coefficient of variation of the seed spacing in wheat sowing with mechanical seed drills was between 100% and 110%. The deviation in the seed spacing was greater than the mean seed spacing. In the wheat sowing at the sowing rate of 300 kg/ha, the mean seed spacing was 20 mm, while the standard deviation of the seed spacing was 22 mm. The authors claimed that one of the main reasons for this non-uniformity was the irregular flow of seeds in the seed tube. The other factors affecting the seed distribution uniformity may be the seed metering device, furrow opener, and furrow conditions. Plain (hollow) seed tubes used in conventional seed drills lead to the unsteady flowing of seeds through the tube, resulting in irregular seed spacing along the row [4,5].

In order to achieve a better seed distribution, the rate of seeds sown at the desired spacing should be increased. Traditional mechanical seed drills drop the seeds in furrow lines in a continuous stream. One of the main reasons for the poor seed distribution in the mechanical seed drills is the irregular movement of the seeds in the seed tube. Delivered (activated) seeds by the metering units and left in the seed tube move in the seed tube completely by chance, under the effect of gravity. Since the number of striking of seeds to the wall of the seed tube and the trajectory of seeds is different, the falling time is also different, and therefore the seed flow is not uniform. As a result, when the seeds leave the seed tube and are placed in the soil, their distribution on the row is irregular (or non-uniform). No matter how good a metering unit is used, the seed flow is disrupted due to the movement of seeds in the seed tube [5,6].

In previous years, extensive efforts have been made by manufacturers to improve volumetric metering devices. However, the problem is that a high uniformity of metering is mixed up in the seed tube, leading again to a random distribution of seed spacing in the field. In order to optimize the performance of a seeder, the structure of the seed-metering device, seed tube, and furrow openers should be optimized in terms of seed movement, trajectory, and deposition [7]. To improve the seed distribution uniformity, Chen and Zhong designed a belt-type seed delivery system for the air-suction metering device. The results of the research showed that the acceptable rate of seed spacing was 98.50% when the height of the outlet (dropping) point of seeds was 100 mm. Through the optimal design of the seed tube, the seeding quality (seed distribution uniformity) was improved. Therefore, the design of the seed tube and the process of seed dropping was found to be the main factor affecting the seeding quality of seeders [8].

Limited research has been carried out to evaluate seed flow uniformity in the seed tubes or seed tube outlet. Endrerud investigated the kinetic energy of seeds sliding and falling down in a seed tube. The author suggested that the seed tube leading to the bend should be inclined at 60° to the horizontal plane [9]. Kumar and Raheman developed an embedded system to detect seed flow in the seed tube and blockages in a seed drill [6]. Müller and Köller used a cascade of V-shaped ducts in the furrow opener to increase the uniformity of seed flow after the seed tube. The seeds sliding in a furrow opener with a V-shaped duct system improved seed spacing uniformity in the row. The authors point out that even for drills utilizing a high-quality seed metering device, the non-optimal layout of tubes can aggravate the seed distribution accuracy in the seed rows [5]. Liu and Yang applied reverse engineering design to design a seed tube of a drill. The authors suggested that reverse engineering for designing the seed tube shortened the design time and reduced design cost compared to theoretical design and experimental study. The built 3D geometric model provides a certain reference for the optimization of seed tubes to improve uniformity

of seed flow [7]. Wang et al. designed a pneumatic wheat-shooting device, aiming to improve the acceleration performance of seeds in seed tubes. The pneumatic wheat-shooting device had a faster and more stable airflow field after the optimization. Results of the field test showed that the optimized pneumatic wheat-shooting device had about 68% higher seeding depth than the initial device [10].

Yazgi et al. evaluated seed trajectories in different seed tubes. They found that the seed release point of the precision metering mechanism to the seed tube is an effective factor in seed distribution uniformity, because the release point of seeds from seed metering mechanisms affects the bouncing of the seeds in the seed tube [11].

Kocher et al. determined the effects of seed tube condition (new or worn) on seed distribution uniformity of the seeders. Corn seeds with round and flat shapes were tested to determine the effect of seed shape on seed flow uniformity in the seed tube. Results showed that seed spacing uniformity was better with the new seed tubes than with the worn seed tubes for all seeders tested in the laboratory with the University of Nebraska Planter Test Stand. Seed spacing uniformity also showed that the round seed used in this experiment had better seed spacing uniformity than the flat seed within the same seed tube condition (new or worn). The authors also suggested that more research on seed tubes is needed to improve seed spacing uniformity [12].

In order to solve the problem of poor uniformity in traditional seed drills, a new helicoidal seed tube (as an attachment to conventional seed tube exit) was developed in this research so that the irregular seed flow in the seed tube can be made more regular just before the seeds fall to the furrow (soil). The regulation of the flow of the seeds by moving on the spiral and reducing the velocity and kinetic energy of falling seeds (by the effect of friction) is aimed at the development of the new helicoidal seed tube. Thus, the preventing displacement of seeds by bouncing and rolling in the furrow and as a result, providing more uniform seed distribution in the row are targeted.

## 2. Materials and Methods

This research was carried out in two-stage laboratory and field trials.

### 2.1. Laboratory Tests

The seed tubes (containing the helix) shown in Figure 1, were made from a PLA+ material with a layer thickness of 0.2 mm using a 3D printer. The dimensions are presented in Table 1. The diameter of the helix in the seed tube was 32 mm. The middle shaft of the helix was 6 mm, and the thickness of the spiral leaf was designed as 0.5 mm.

Laboratory trials were repeated for the following dimensions of the helix.

- Three heights of helix (10, 15, and 20 cm) and
- Four pitch sizes (28, 32, 36, and 40 mm)

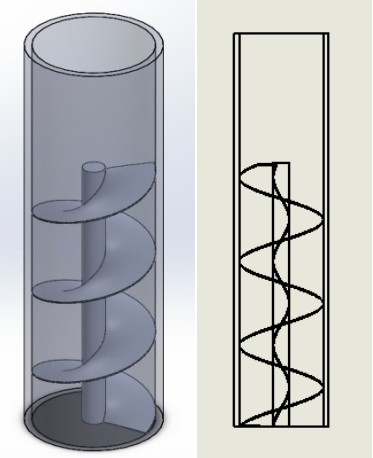

**Figure 1.** Helicoidal seed tube.

**Table 1.** Helicoidal seed tube dimensions subjected to laboratory tests.

| Height of Helix (mm) | 100 | 150 | 200 |
|---|---|---|---|
| Pitch size (mm) | 28 | 28 | 28 |
| | 32 | 32 | 32 |
| | 36 | 36 | 36 |
| | 40 | 40 | 40 |

The twelve different helicoidal seed tube with the dimensions in Table 1 was manufactured and then mounted on the tip of the seed tube of a seed drill, as shown in Figure 2 for laboratory trials. Wheat and barley seeds were used in the experiments. The main dimensions of the seeds are presented in Table 2.

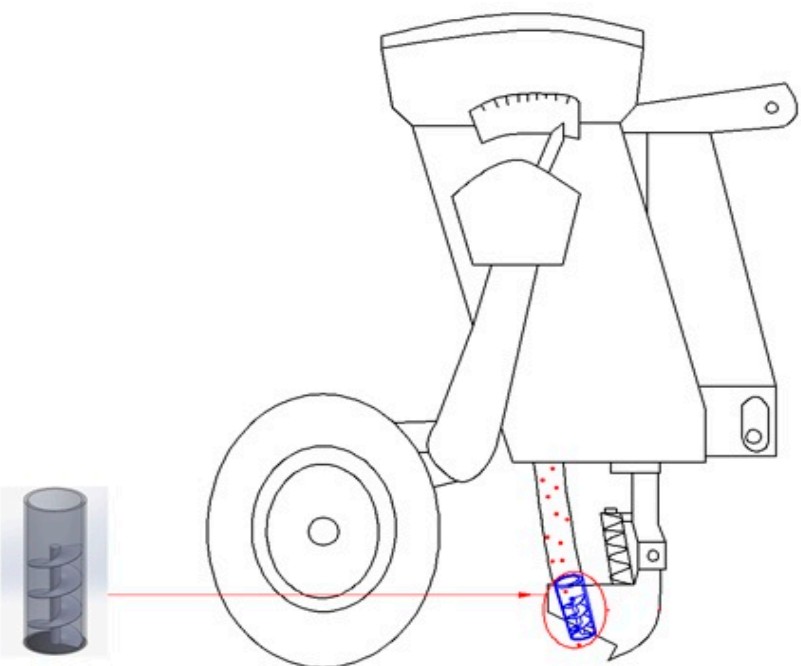

**Figure 2.** The seed drill to which the developed helicoidal seed tube was mounted during the evaluation.

**Table 2.** The main dimensions of wheat and barley seeds used in the study.

| Seed | Seed Length (mm) | Seed Width (mm) | Seed Thickness (mm) | One Thousand Kernel Mass (g) |
|---|---|---|---|---|
| Wheat | 7.3 | 3.3 | 2.8 | 47.2 |
| Barley | 7.4 | 2.7 | 1.9 | 43.5 |

In the laboratory experiments, the falling speed of the seeds and the time between successive seeds from the seed tube exit were determined with the aid of a high-speed camera system following the method developed by Karayel et al. [13]. The high-speed camera system included three main components: a high-speed digital camera for recording of passing seeds, software for image processing, and a computer for data processing and monitoring. In order to obtain acceptable resolutions of the seeds, a frame rate of 750 frames/s was chosen for this research.

In the laboratory and field trials, a Mert-San model M2000 (Mert-San, Tekirdag, Turkey) seed drill with a fluted feed roller type seed metering unit and shoe type furrow openers was used (Figure 3). In the laboratory trials, the seed drill was suspended, and the wheel of the seed drill was rotated at a desired speed by using the test stand shown in Figure 4. A high-speed camera was positioned across the seed tube at a distance of 400 mm to record the seed flow.

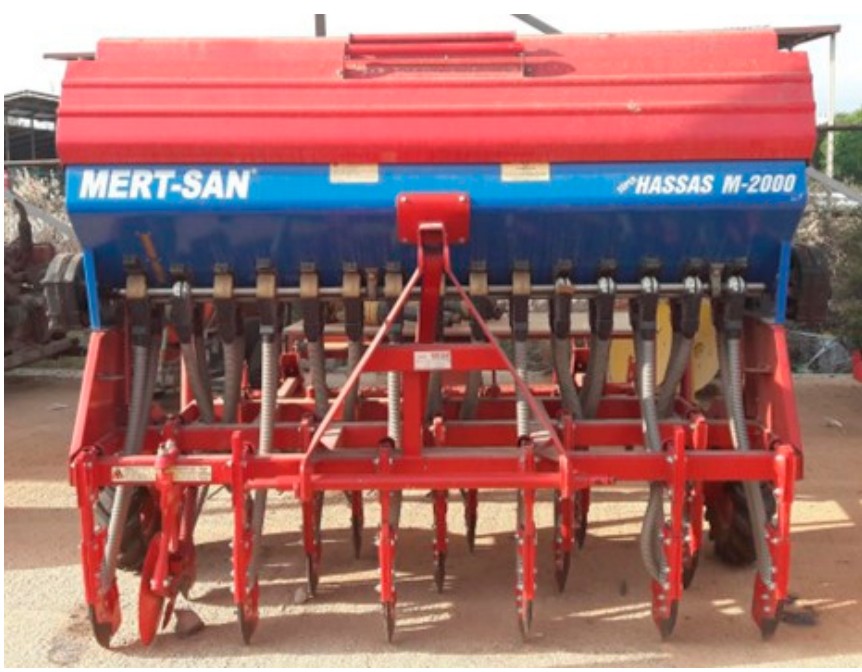

**Figure 3.** Seed drill used in laboratory and field tests.

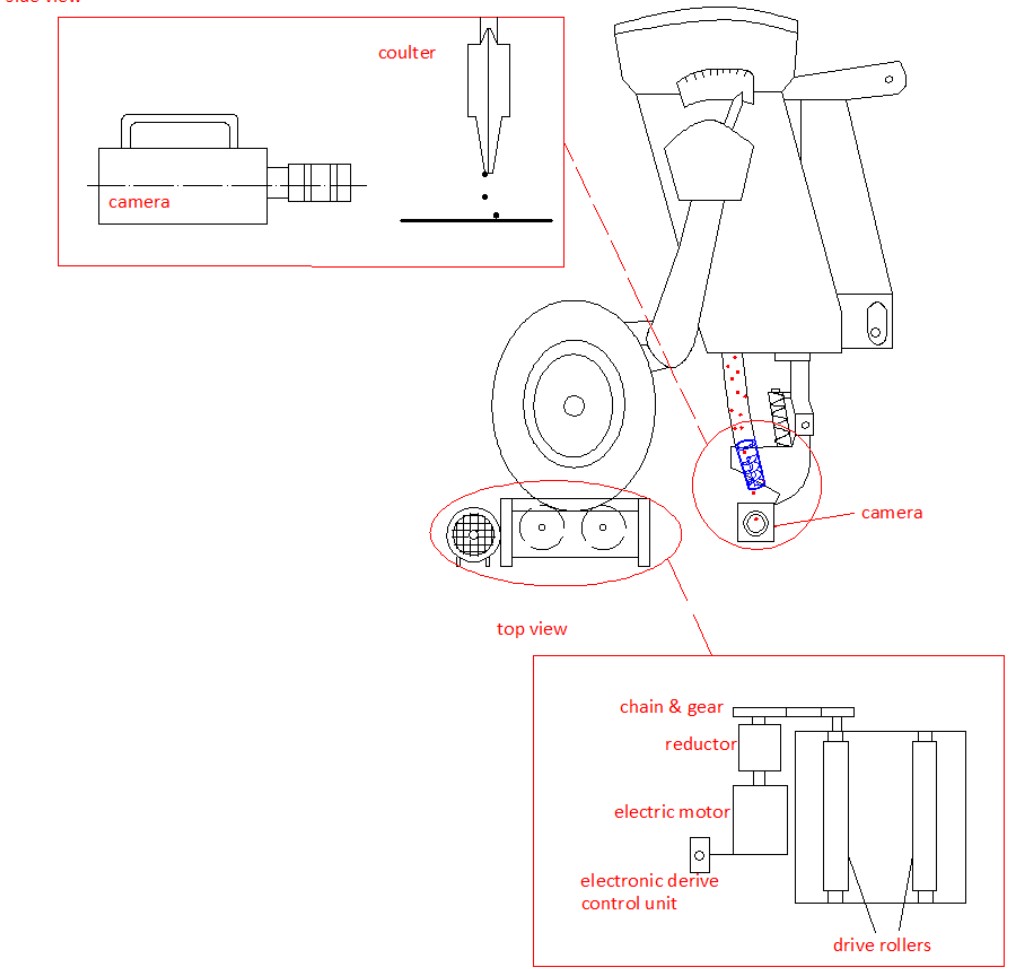

**Figure 4.** Seed drill suspended on a test stand during the laboratory testing.

*2.2. Field Tests*

　　　Field trials were conducted in loamy soil at the Research and Application Farm of Agriculture Faculty on the Campus of Akdeniz University. The seedbed was thoroughly prepared by plowing to an approximate depth of 30–35 cm, disc harrowing, and leveling. The workable soil had no rocks or hard clay clods and was without crop residues. Tests were conducted at near-optimum soil moisture for tillage and sowing. The seed drill was operated over the seedbed at a forward speed of 1.0 m/s and a seeding rate of 200 kg/ha.

　　　The seed distribution uniformity was measured by allowing sowed seeds to emerge after 20 days, and the distances between successive plants (seed spacing) along the rows were measured. The mean, standard deviation, and coefficient of variation of the measured seed spacings were calculated.

　　　Seed distribution in the horizontal plane relative to the direction of the seeder should be specified by seed spacing and lateral seed scatter of seed placement [14]. Therefore, the lateral seed scatter (deviation from row center of seeds) was also measured.

　　　The plant growing area and uniformity of growing area distribution for each plant were analysed using the methods described in Karayel and Karayel et al. [15,16]. The plant growing area was calculated using fallowing m-file written in MATLAB software: [v,c] = voronoin(x), for j = 1: length(c); A = polyarea(v(c{j},1), v(c{j},2)), end.

　　　Each experiment was replicated five times. An analysis of variance method was applied to analyze the velocity of fall of seeds and time between successive seeds obtained from computer-aided motion analysis. Duncan's multiple-range tests were used to identify significantly different means within dependent variables. An independent sample *t*-test was applied to compare data sets (including seed spacings, the velocity of falling seeds, and plant growing area) obtained from laboratory and field trials. Analysis of variance or *t*-test was not applied on the standard deviation and coefficient of variation.

## 3. Results and Discussion

　　　The velocity of falling seeds and the times between successive seeds as determined in laboratory tests using the high-speed camera are presented in Tables 3 and 4.

　　　The effects of the helix height and pitch size on the velocity of falling seeds were not statistically significant. As the helix height increased, the standard deviation and coefficient of variation of the velocity of falling seeds decreased (Table 5). Increasing the helix height improved seed flow uniformity. However, some blockage in helicoidal section of seed tube were observed when the helix was higher than 200 mm. Therefore, the height of helix was limited to 200 mm.

**Table 3.** The velocity of falling seed in the helicoidal seed tube measured in laboratory tests as influenced by the height of the helix and pitch size.

| Height of Helix (mm) | Pitch Size (mm) | Velocity of Fall of Seeds (m/s) | | Standard Deviation of Velocity of Fall of Seeds (cm) | | Coefficient of Variation of Velocity of Fall of Seeds (%) | |
|---|---|---|---|---|---|---|---|
| | | **Wheat** | **Barley** | **Wheat** | **Barley** | **Wheat** | **Barley** |
| 100 | 28 | 0.47 | 0.46 | 0.072 | 0.079 | 15.8 | 17.17 |
| | 32 | 0.56 | 0.49 | 0.058 | 0.077 | 10.46 | 15.71 |
| | 36 | 0.59 | 0.55 | 0.058 | 0.071 | 9.83 | 12.91 |
| | 40 | 0.65 | 0.62 | 0.071 | 0.070 | 10.9 | 11.29 |
| 150 | 28 | 0.65 | 0.58 | 0.093 | 0.088 | 14.30 | 15.17 |
| | 32 | 0.51 | 0.62 | 0.064 | 0.060 | 12.54 | 9.67 |
| | 36 | 0.65 | 0.64 | 0.068 | 0.062 | 10.46 | 9.68 |
| | 40 | 0.69 | 0.66 | 0.074 | 0.074 | 10.72 | 11.21 |
| 200 | 28 | 0.68 | 0.62 | 0.062 | 0.055 | 9.11 | 8.87 |
| | 32 | 0.65 | 0.65 | 0.043 | 0.045 | 6.66 | 6.92 |
| | 36 | 0.59 | 0.66 | 0.025 | 0.032 | 4.54 | 4.24 |
| | 40 | 0.66 | 0.69 | 0.042 | 0.044 | 6.36 | 6.38 |

**Table 4.** The mean time, standard deviation, and coefficient of variation between successive seeds in the helicoidal seed tube measured in laboratory tests.

| Height of Helix (mm) | Pitch Size (mm) | Mean Time between Successive Seeds (s) | | Standard Deviation of Time between Successive Seeds (s) | | Coefficient of Variation of Time between Successive Seeds (%) | |
|---|---|---|---|---|---|---|---|
| | | Wheat | Barley | Wheat | Barley | Wheat | Barley |
| 100 | 28 | 0.024 | 0.026 | 0.0067 | 0.0074 | 27.92 | 28.46 |
| | 32 | 0.023 | 0.025 | 0.0061 | 0.0064 | 26.52 | 25.60 |
| | 36 | 0.022 | 0.025 | 0.0043 | 0.0042 | 19.55 | 16.85 |
| | 40 | 0.022 | 0.025 | 0.0061 | 0.0063 | 27.73 | 25.20 |
| 150 | 28 | 0.029 | 0.028 | 0.0077 | 0.0071 | 26.55 | 25.35 |
| | 32 | 0.033 | 0.029 | 0.0066 | 0.0063 | 20.01 | 21.72 |
| | 36 | 0.025 | 0.027 | 0.0052 | 0.0059 | 20.80 | 21.85 |
| | 40 | 0.033 | 0.029 | 0.0071 | 0.0069 | 21.52 | 23.79 |
| 200 | 28 | 0.035 | 0.033 | 0.0077 | 0.0053 | 22.02 | 16.06 |
| | 32 | 0.031 | 0.033 | 0.0060 | 0.0043 | 19.35 | 13.03 |
| | 36 | 0.029 | 0.030 | 0.0032 | 0.0023 | 11.03 | 7.67 |
| | 40 | 0.032 | 0.034 | 0.0068 | 0.0055 | 21.25 | 16.18 |

**Table 5.** The effect of the height of helix and pitch size of helicoidal seed tube on the mean velocity of fall of seeds, standard deviation, and coefficient of variation of velocity of fall of seeds, according to results of laboratory tests.

| Height of Helix (mm) | Mean Velocity of Fall of Seeds (m/s) | | Standard Deviation of Velocity of Fall of Seeds (cm) | | Coefficient of Variation of Velocity of Fall of Seeds (%) | |
|---|---|---|---|---|---|---|
| | Wheat | Barley | Wheat | Barley | Wheat | Barley |
| 100 | 0.57 | 0.55 | 0.06 | 0.08 | 11.75 | 14.28 |
| 150 | 0.63 | 0.63 | 0.07 | 0.07 | 12.43 | 11.51 |
| 200 | 0.64 | 0.66 | 0.04 | 0.04 | 6.67 | 6.75 |
| Significance | NS | NS | | | | |
| Pitch size (mm) | | | | | | |
| 28 | 0.60 | 0.55 | 0.08 | 0.07 | 13.07 | 13.74 |
| 32 | 0.58 | 0.58 | 0.06 | 0.06 | 10.68 | 11.51 |
| 36 | 0.60 | 0.62 | 0.05 | 0.06 | 8.28 | 9.14 |
| 40 | 0.67 | 0.66 | 0.06 | 0.06 | 9.33 | 9.63 |
| Significance | NS | NS | | | | |

NS, non-significant at probability; $p < 0.05$.

The effects of helix height and pitch size of helicoidal seed tube on the time between successive seeds were not statistically significant. In terms of standard deviation and coefficients of variation, the height of the helicoidal seed tube was the more effective factor than the pitch size. The lowest standard deviation and coefficient of variation of time between successive seeds were obtained at 200 mm helix height (Table 6).

According to Tables 3–6, the lowest standard deviation and the coefficient of variation of the time between successive seeds were obtained when the helicoidal seed tube with a helix height of 200 mm and pitch size of 36 mm was used. For this reason, the helicoidal seed tube with a length of 200 mm and pitch size of 36 mm was subjected to laboratory and field tests to compare with conventional seed tubes.

Mean, standard deviation and coefficient of variation of seed spacings obtained in the laboratory tests are presented in Table 7 to compare seed tubes. Statistical analysis (*t*-test) was performed to compare the mean seed spacings of helicoidal and conventional seed tubes for each seed. According to the result of the statistical test, the effect of seed tube type on seed spacing was significant. While the mean seed spacing was 2.99 cm and 2.83 cm for conventional seed tubes, this value has decreased to 1.96 cm and 2.02 cm for the

helicoidal seed tube for wheat and barley seed, respectively. The laboratory test showed that using the helicoidal seed tube decreased the coefficient of variation of the seed spacing from 168.13% and 148.76% to 79.02% and 75.24% for wheat and barley seeds, respectively.

**Table 6.** The effect of the height of helix and pitch size of helicoidal seed tube on the time between successive seeds, standard deviation and coefficient of variation of time between successive seeds, according to results of laboratory tests.

| Height of Helix (mm) | Mean Time between Successive Seeds (s) | | Standard Deviation of Time between Successive Seeds (s) | | Coefficient of Variation of Time between Successive Seeds (%) | |
|---|---|---|---|---|---|---|
| | **Wheat** | **Barley** | **Wheat** | **Barley** | **Wheat** | **Barley** |
| 100 | 0.024 | 0.025 | 0.007 | 0.006 | 25.43 | 24.02 |
| 150 | 0.030 | 0.028 | 0.007 | 0.007 | 22.22 | 23.17 |
| 200 | 0.032 | 0.032 | 0.006 | 0.004 | 18.41 | 13.23 |
| Significance | NS | NS | | | | |
| Pitch size (mm) | | | | | | |
| 28 | 0.029 | 0.029 | 0.007 | 0.007 | 25.49 | 23.29 |
| 32 | 0.029 | 0.029 | 0.007 | 0.006 | 22.84 | 20.91 |
| 36 | 0.025 | 0.027 | 0.004 | 0.004 | 17.12 | 15.45 |
| 40 | 0.029 | 0.029 | 0.007 | 0.006 | 23.50 | 21.72 |
| Significance | NS | NS | | | | |

NS, non-significant at probability; $p < 0.05$.

**Table 7.** The influence of seed tubes on seed distribution uniformity, as indicated by the mean seed spacing and its corresponding standard deviation and coefficient of variation as tested in the laboratory.

| Seed Tube | Mean Seed Spacing (cm) | | Standard Deviation of Seed Spacing (cm) | | Coefficient of Variation of Seed Spacing (%) | |
|---|---|---|---|---|---|---|
| | **Wheat** | **Barley** | **Wheat** | **Barley** | **Wheat** | **Barley** |
| Conventional (hollow seed tube) | 2.99 | 2.83 | 5.02 | 4.21 | 168.13 | 148.76 |
| Helicoidal seed tube | 1.96 | 2.02 | 1.55 | 1.52 | 79.02 | 75.24 |

The effect of seed tube type on the velocity of falling seeds was statistically significant for both seeds, as shown in Table 8. The mean velocity of falling wheat seeds in the conventional seed tube was 2.96 m/s, which was observed to decrease at a velocity of 0.62 m/s in the helicoidal seed tube. Similarly, the standard deviation and coefficient of variation of the velocity of falling wheat seeds were 0.77 m/s and 26.17% for conventional seed tubes, it decreased to 0.05 m/s and 8.06% for helicoidal seed tubes, respectively. Similar results were also obtained in the experiments with barley seeds.

**Table 8.** The effect of seed tubes on the velocity of falling seeds.

| Seed Tube | Mean Velocity of Falling Seeds (m/s) | | Standard Deviation of Velocity of Falling Seeds (m/s) | | Coefficient of Variation of Falling Seeds (%) | |
|---|---|---|---|---|---|---|
| | **Wheat** | **Barley** | **Wheat** | **Barley** | **Wheat** | **Barley** |
| Conventional (hollow seed tube) | 2.96 | 2.88 | 0.77 | 0.69 | 26.17 | 23.95 |
| Helicoidal seed tube | 0.62 | 0.66 | 0.05 | 0.06 | 8.06 | 8.81 |

Lei et al. found the phenomenon of collision and bounce between seeds and tube wall and this changed seed motion trajectory with the length of the seed tube, influencing the uniformity of seed-dropping time. The author suggested that the uniformity of seed flow in the seed tube could be improved by reducing the number of collisions in the seed tube, thus, the variation in seed velocity [17]. The helicoidal seed tube used in our experiments

regulated seed flow, reduced the number of collisions in the seed tube, and the variation in seed velocity, and thus, increased seed distribution uniformity.

The velocity of the fall of a seed is a contributing factor to the bouncing and rolling of the seeds in the furrow [7]. Therefore, the falling velocity of the seed should be minimum in order to lessen bouncing and rolling to avoid seed displacement out of the furrow. In addition, the coefficient of variation of the time between successive seeds is an important indicator of seed distribution uniformity in the furrow. Seed spacings will be formed by multiplying the time between successive seeds by the forward speed of the seed drill. Therefore, a lower coefficient of variation of time between successive seeds is required for a better seed distribution uniformity.

Our results support reports from Müller and Köller, who found the positive effect of V-shaped ducts placed in the furrow opener on the uniformity of seed flow. The friction of the falling seeds while sliding on ducts decreased kinetic energy of the seeds and therefore prevented the bouncing or rolling in the furrow [4,5]. A similar effect was observed during the sliding of seeds on the screw of the helix in this study.

According to the results of the laboratory trials, the seed flow with the helicoidal seed tube has become more regular than with the conventional seed tube. Thus, the seed's distribution accuracy has been improved. The helicoidal seed tube also reduced the mean velocity of falling seeds, allowing the seeds to fall to the furrow at lower speeds.

The mean, standard deviation, and coefficient of variation of seed spacings obtained in the field tests are presented in Table 9. The effect of seed tube type on seed spacing was statistically significant. While the mean spacing of wheat seeds was 3.7 cm for the conventional seed tube, this value has decreased to 2.5 cm for the helicoidal seed tube. While the standard deviation and coefficient of variation of spacings between wheat seeds were 4.4 cm and 118.4% for conventional seed tubes, it decreased to 1.9 cm and 77.2% for helicoidal seed tubes, respectively. The positive effect of the helicoidal seed tube on the seed distribution uniformity in the field was also observed in the experiments with barley seeds.

**Table 9.** The effect of seed tubes on the seed's distribution accuracy in field trials.

| Seed Tube | Mean Seed Spacing (cm) | | Standard Deviation of Seed Spacing (cm) | | Coefficient of Variation of Seed Spacing (%) | | Mean Lateral Seed Scatter (cm) | |
|---|---|---|---|---|---|---|---|---|
| | Wheat | Barley | Wheat | Barley | Wheat | Barley | Wheat | Barley |
| Conventional (hollow seed tube) | 3.7 | 3.0 | 4.4 | 4.2 | 118.4 | 139.5 | 1.5 | 1.6 |
| Helicoidal seed tube | 2.5 | 2.6 | 1.9 | 1.8 | 77.2 | 70.6 | 0.9 | 0.8 |

The lateral seed scatter (the deviation from the row center of the seeds) was lower when the helicoidal seed tube was used. The lower the velocity of falling seeds to the furrow, the reduced the lateral seed scattering.

The effect of conventional and helicoidal seed tubes on the plant growing area is presented in Table 10. The effect of seed tube type on plant growing area was statistically significant. The mean, standard deviation, and coefficient of variation of plant growing area decreased when the helicoidal seed tube was used.

The helicoidal seed tube provided a more equal growing area for each plant (Figure 5). Thus, competition between plants will be reduced and they will be provided to benefit equally from factors such as soil moisture, plant nutrients, and sunlight.

**Table 10.** The effect of seed tubes on plant growing area.

| Seed Tube | Seed | Mean Plant Growing Area (cm$^2$) | Standard Deviation of Plant Growing Area (cm$^2$) | Coefficient of Variation of Plant Growing Area (%) |
|---|---|---|---|---|
| Conventional (hollow seed tube) | Wheat | 60.83 | 55.13 | 91 |
| | Barley | 58.24 | 52.48 | 90 |
| Helicoidal seed tube | Wheat | 39.82 | 23.53 | 59 |
| | Barley | 37.98 | 21.60 | 57 |

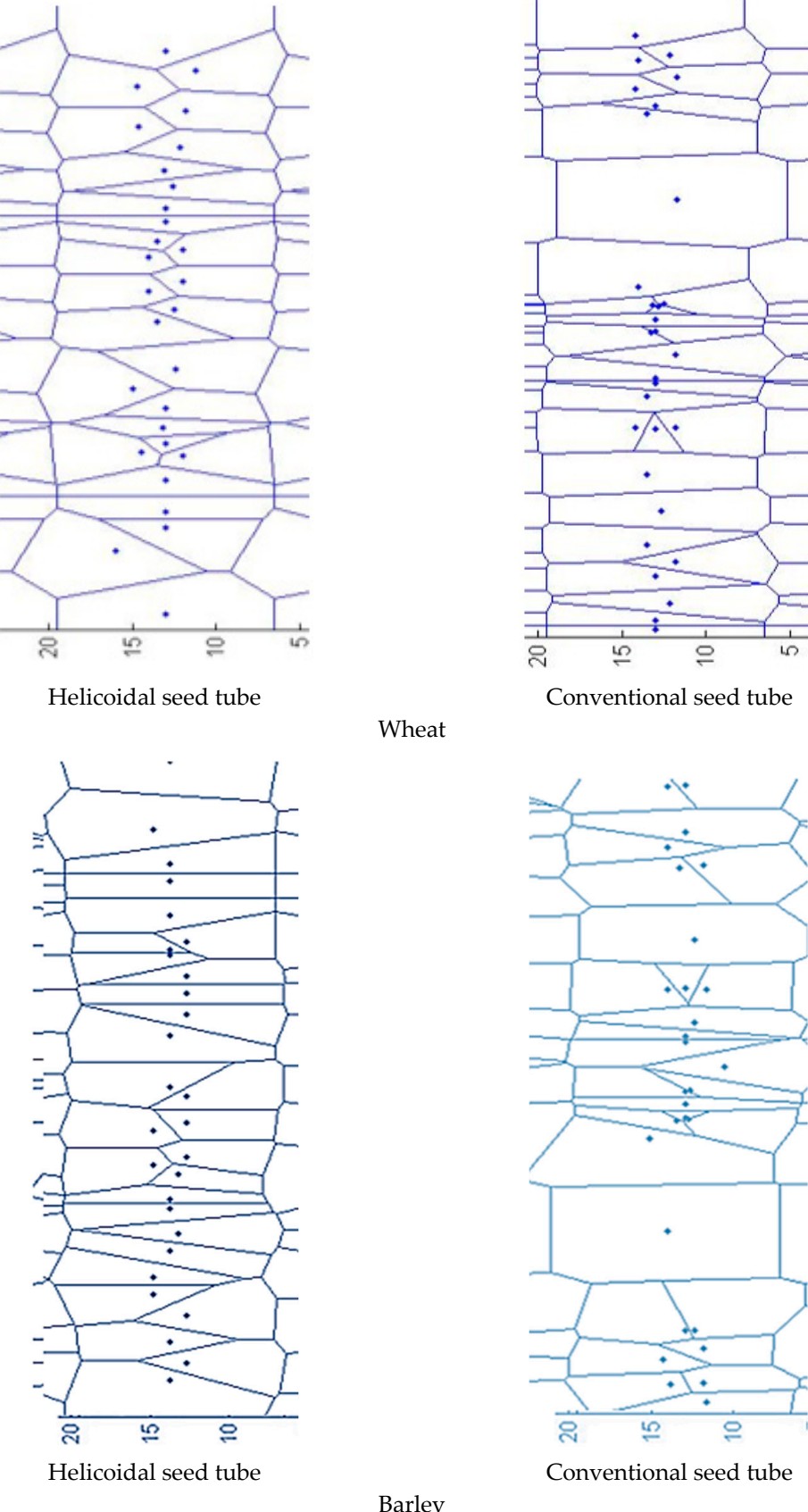

**Figure 5.** Growing area distribution for each plant.

## 4. Conclusions

A new type of seed tube called a helicoidal seed tube was designed. The optimum value of helix height and pitch size of the seed tube was determined as 200 mm and 36 mm, respectively, in the laboratory. Then, the performance of the helicoidal seed tube in terms of seed distribution uniformity was compared with the conventional seed tube for laboratory and field conditions.

The seed drill was operated at the seeding rate of 200 kg/ha and forward speed of 1 m/s both at the laboratory and field trials. The use of a helicoidal seed tube reduced the mean, standard deviation, coefficient of variation of seed spacings and velocity of falling seeds as compared to the conventional seed tube.

The process of seed ejection from the seed tube is one of the main factors affecting the seeding performance of seed drills. The helicoidal seed tube reduced the number of collisions in the seed tube and the variation in seed velocity. Allowed the seeds to fall to the furrow with lower velocity and kinetic energy. The low kinetic energy of the seeds reduced their displacement by bouncing or rolling.

As a result, the helicoidal seed tube provided a more equal distribution of the seed and the growing area per plant in the field. This will contribute to reducing competition between plants.

## 5. Patents

The helicoidal seed tube in this paper, as indicated in Figures 1 and 2, used for the experimental analysis is protected under Turkey Republic Law (Patent Number: 2018-10225, date of registration: 22 February 2021).

**Author Contributions:** Conceptualization, formal analysis, investigation, methodology, validation, writing—original draft preparation, D.K.; methodology, validation, supervision, writing—original draft preparation, E.Š.; conceptualization, methodology, investigation, A.A. All authors have read and agreed to the published version of the manuscript.

**Funding:** This research received no external funding.

**Conflicts of Interest:** The authors declare no conflict of interest.

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
