# Peer review of "Design and Experiment of a Helicoidal Seed Tube to Improve Seed Distribution Uniformity of Seed Drills"

_processes, doi:10.3390/pr10071271_

Round 1

Reviewer 1 Report

A new type seed tube named as helicoidal seed tube was designed in this study. The use of helicoidal seed tube reduced the mean, standard deviation, coefficient of variation of seed spacings and velocity of falling seeds as compared to the conventional seed tube. The proposed design is innovative, but there are some issues that require explanation and clarification.

1. Abstract: “In the conventional seed tube of a drill seeder, the flow of seeds into the planting bed is usually disrupted and irregular which contribute to a poor seed distribution efficiency.” What are the main reasons for the disrupted and irregular flow of seeds? Will the performance of the seed metering device affect it? How to ensure that only seed tube effects are considered?

2. Introduction: “The rate of seeds sown at the desired spacing over the row was very low for mechanical seed drills. The coefficient of variation of the seed spacing in wheat sowing with mechanical seed drills was between 100% and 110%. The deviation in the seed spacing was greater than the mean seed spacing. In the wheat sowing at the sowing rate of 300 kg/ha, the mean seed spacing was 20 mm, while the standard deviation of the seed spacing was 22 mm.” Are the relevant data supported by references or documents? It is recommended to supplement relevant references.

3. “The authors claimed that the main reason for this non-uniformity was the irregular flow of seeds in the seed tube. Plain (hollow) seed tubes used in conventional seed drills leads to unsteady flowing of seeds through the tube, resulting in irregular seed spacing along the row.”  “One of the main reasons for the poor seed distribution in the mechanical seed drills is the irregular movement of the seeds in the seed tube.” The main reason or one of the main reasons needs to be clear and unified. What are the other reasons? 

4. “Xia et al. developed a no-tube seeding method for wheat sowing to optimize ground clearance of seed drills [11]. Because the authors found that the ground clearance with the seeding accuracy and uniformity were both quadratic functions. The no-tube seed drill was composed of rotary tillage mechanism, slotting device, press wheel, seed and fertilizer boxes, soil covering cutters, furrow openers, hydraulic lift. The field tests showed that when ground clearance was 7 cm, the sowing accuracy of no-tube seed drill was 83.84% and the variation coefficient of seeding uniformity was 14.68%. All results were significantly superior to traditional seed drill.” Is the reference appropriate? The absence of a seed tube is not a concern of this study, and the design and structure of the seed tube are obviously more important.

5. “The regulation of the flow of the seeds by moving on the spiral, and Reducing velocity and kinetic energy of falling seeds (by the effect of friction). Thus, the preventing displacement of seeds by bouncing and rolling in the furrow, and as a result, providing more uniform seed distribution on the row.” Reduce the speed of seed falling, and the movement time of seeds in the seed tube will increase. Will this approach affect the speed of the seed metering device and the forward speed of the seeder?

6. “The seed tubes (contain the helix) shown in Figure 1 were made from a PLA+ material with a layer thickness of 0.2 mm using a 3D printer. The dimensions given in Table 1. The diameter of helix in seed tube was 32 mm. The middle shaft of the helix was 6 mm, and the thickness of the spiral leaf is designed as 0.5 mm.” How are the relevant parameters determined? Such as: The diameter of helix in seed tube was 32 mm. Why?

7. ”In the laboratory experiments, the falling speed of the seeds and the time between successive seeds from the seed tube exit were determined with the aid of a high-speed camera following the method developed by Karayel et al. [13].” It is recommended to briefly introduce the content of the method.

Author Response

Reviewer1

Thank you very much for your valuable comments. Your comments are quite useful to improve the paper. We have improved the manuscript and made further changes accordingly. We hope these changes will meet with your approval. These changes are listed below.

  1. Abstract: “In the conventional seed tube of a drill seeder, the flow of seeds into the planting bed is usually disrupted and irregular which contribute to a poor seed distribution efficiency.” What are the main reasons for the disrupted and irregular flow of seeds? Will the performance of the seed metering device affect it? How to ensure that only seed tube effects are considered?

Response: We agree with the reviewer that the seed tube of seed drill is not the only reason of the disrupted and irregular seed flow. The first sentence of abstract has been changed and this situation has been mentioned in lines between 10-12

  1. Introduction: “The rate of seeds sown at the desired spacing over the row was very low for mechanical seed drills. The coefficient of variation of the seed spacing in wheat sowing with mechanical seed drills was between 100% and 110%. The deviation in the seed spacing was greater than the mean seed spacing. In the wheat sowing at the sowing rate of 300 kg/ha, the mean seed spacing was 20 mm, while the standard deviation of the seed spacing was 22 mm.” Are the relevant data supported by references or documents? It is recommended to supplement relevant references.

Response: We have checked the related references again. The following references cited in line 63 supports the data.

Müller, J.; Köller, K. Improvement of seed spacing for seed drills. AgEng '96 International Conference on Agricultural Engineering', Madrid, 23-26 September 1996, pp.323-324, Paper 96A-030

Müller, J.; Köller, K. Untersuchung eines drillschars mit integrierter V-profilrolle zur verleichmaBigung der kornlanhsabstande bei der drillsaat (Investigation of a disc coulter with integrated V-Shape coil to improve seed spacing). Agrartechnische Forschung 1996, 2(2), 94-101

  1. “The authors claimed that the main reason for this non-uniformity was the irregular flow of seeds in the seed tube. Plain (hollow) seed tubes used in conventional seed drills leads to unsteady flowing of seeds through the tube, resulting in irregular seed spacing along the row.”  “One of the main reasons for the poor seed distribution in the mechanical seed drills is the irregular movement of the seeds in the seed tube.” The main reason or one of the main reasons needs to be clear and unified. What are the other reasons? 

Response: The seed tube is one of the main reasons of the poor seed distribution. The related sentence has been corrected (line 58).

The other factors affecting the seed distribution uniformity may be the seed metering device, furrow opener and furrow conditions. (lines 59-61)

  1. “Xia et al. developed a no-tube seeding method for wheat sowing to optimize ground clearance of seed drills [11]. Because the authors found that the ground clearance with the seeding accuracy and uniformity were both quadratic functions. The no-tube seed drill was composed of rotary tillage mechanism, slotting device, press wheel, seed and fertilizer boxes, soil covering cutters, furrow openers, hydraulic lift. The field tests showed that when ground clearance was 7 cm, the sowing accuracy of no-tube seed drill was 83.84% and the variation coefficient of seeding uniformity was 14.68%. All results were significantly superior to traditional seed drill.” Is the reference appropriate? The absence of a seed tube is not a concern of this study, and the design and structure of the seed tube are obviously more important.

Response: After this comment, we think that this reference is not relevant to the contents of the manuscript. Therefore, we have removed it.

  1. “The regulation of the flow of the seeds by moving on the spiral, and Reducing velocity and kinetic energy of falling seeds (by the effect of friction). Thus, the preventing displacement of seeds by bouncing and rolling in the furrow, and as a result, providing more uniform seed distribution on the row.” Reduce the speed of seed falling, and the movement time of seeds in the seed tube will increase. Will this approach affect the speed of the seed metering device and the forward speed of the seeder?

Response: This approach will probably affect the speed of the seed metering device and the forward speed of the seed drill. We have not tested it, yet. We are planning the evaluate the seed tube for different speed of metering device and forward speed in our future study. Thank you very much for this significant comment and recommendation.

  1. “The seed tubes (contain the helix) shown in Figure 1 were made from a PLA+ material with a layer thickness of 0.2 mm using a 3D printer. The dimensions given in Table 1. The diameter of helix in seed tube was 32 mm. The middle shaft of the helix was 6 mm, and the thickness of the spiral leaf is designed as 0.5 mm.” How are the relevant parameters determined? Such as: The diameter of helix in seed tube was 32 mm. Why?

Response: The helicoidal seed tube was mounted at the tip of the conventional seed tube (hollow seed tube) of seed drill, so the diameter of both seed tube must be the equal (line 144).

Other dimensions such as middle shaft of the helix and thickness of the spiral leaf were determined by considering the greater seed flow volume.

  1. ”In the laboratory experiments, the falling speed of the seeds and the time between successive seeds from the seed tube exit were determined with the aid of a high-speed camera following the method developed by Karayel et al. [13].” It is recommended to briefly introduce the content of the method.

Response: A brief introduces has been added (lines 154-158)

Reviewer 2 Report

The manuscript entitles “Design and Experiment of a Helicoidal Seed Tube to Improve Seed Distribution Uniformity of Seed Drills” has been written very well and have some comments below:

In Keywords section, Keywords should be different from the title.

Line no 32; In Introduction part, Please add some references as you have written in first paragraph.

Line no 62-72; Please add some references.

Line no 141. Sentence should be re-corrected like Yazgi et al. [10].

Line no 130-131. Remove the bullets as you have used in your objectives and write in a consecutive one sentence.

Line no 148. Figure 1 use full stop after the legends.

Figure 3 should be in high quality JPEG image.

In methodology part, authors need to provide information about how many replication they used for this experiment in lab and field.

How did the authors find results are significantly proved? There is not any inside the tbales as mentioned below the tables that Means within a group followed by same letter are not significantly different at probability 223 P<0.05, by Duncan’s multiple range test.

References should be according to the journals guideline. And follow same format for all the references.

English language must be improved in throughout the manuscript.

Author Response

Reviewer2

Thank you very much for your valuable comments. Your comments are quite useful to improve the paper. We have improved the manuscript and made further changes accordingly. We hope these changes will meet with your approval. These changes are listed below.

In Keywords section, Keywords should be different from the title.

Response: Some keywords same as title have been changed and new ones have been added

Line no 32; In Introduction part, Please add some references as you have written in first paragraph.

Response: References have been added (line 46)

Line no 62-72; Please add some references.

Response: References have been added (line 74)

Line no 141. Sentence should be re-corrected like Yazgi et al. [10].

Response: We indicated the number of references at the end of the sentence as following because of the procedures in the journals guideline.

Yazgi et al. compared seed trajectories in different seed tubes. They found that the seed release point of precision metering mechanism to seed tube was an effective factor on seed distribution uniformity [10].

Some corrections on sentence have been done (lines 106, 107, 109).

Line no 130-131. Remove the bullets as you have used in your objectives and write in a consecutive one sentence.

Response: The bullets have been removed and objectives have been written in a consecutive one sentence (lines 122-126)

Line no 148. Figure 1 use full stop after the legends.

Response: Full stop have been added (line 140)

Figure 3 should be in high quality JPEG image.

Response: Figure 3 has been replaced with a higher quality JPEG image (Page 5).

In methodology part, authors need to provide information about how many replication they used for this experiment in lab and field.

Response: The information about replication have been added (line 189).

How did the authors find results are significantly proved? There is not any inside the tables as mentioned below the tables that Means within a group followed by same letter are not significantly different at probability 223 P<0.05, by Duncan’s multiple range test.

Response: Thanks for the reviews' advice. We are sorry for making the mistake. The lines include explanation of “Means within a group followed by same letter are not significantly different at probability 223 P<0.05, by Duncan’s multiple range test” have been removed.

References should be according to the journals guideline. And follow same format for all the references.

Response: The reference style has been checked according to journal guideline.

English language must be improved in throughout the manuscript.

Response: The English language of manuscript have been improved throughout the manuscript.
